# Cytoplasmic Clusterin Suppresses Lung Cancer Metastasis by Inhibiting the ROCK1-ERK Axis

**DOI:** 10.3390/cancers14102463

**Published:** 2022-05-17

**Authors:** Shaobo Huang, Xu Li, Weiqi Gu, Xiaoyi Li, Jingjing Zhao, Jueheng Wu, Junchao Cai, Xianming Feng, Tianyu Tao

**Affiliations:** 1Department of Microbiology, Zhongshan School of Medicine, Sun Yat-Sen University, Guangzhou 510080, China; huangshb7@mail2.sysu.edu.cn (S.H.); lixu26@mail2.sysu.edu.cn (X.L.); guwq3@mail2.sysu.edu.cn (W.G.); lixy395@mail2.sysu.edu.cn (X.L.); wujh@mail.sysu.edu.cn (J.W.); 2Department of Cardiac Surgery Center, The First Affiliated Hospital of Sun Yat-sen University, Guangzhou 510080, China; zhao833@163.com; 3Department of Immunology, Sun Yat-sen University Zhongshan School of Medicine, Guangzhou 510080, China; caijch3@mail.sysu.edu.cn; 4Department of Gastrointestinal Surgery Center, The First Affiliated Hospital of Sun Yat-sen University, Guangzhou 510060, China

**Keywords:** Clusterin, ROCK1, metastasis, lung cancer

## Abstract

**Simple Summary:**

We show that CLU, especially cytoplasmic precursor CLU, is downregulated in lung cancer and correlates with poor survival. The silencing of CLU promotes lung cancer cell migration and invasion, while the overexpression of CLU potently inhibits these phenomena. Interestingly, secretory CLU proteins are slightly decreased in lung cancer tissue and fail to exert similar anti-metastatic effects like cytoplasmic precursor CLU, demonstrating that cytoplasmic precursor CLU is the primary functional isoform of CLU, which exerts the anti-metastatic effects of lung cancer. Mechanistically, cytoplasmic precursor CLU binds ROCK1 to decrease phosphorylation of ERK1/2 by inhibiting the kinase activity of ROCK1, leading to an anti-metastatic effect in lung cancer cells. These findings reveal a novel insight into the function and regulation of cytoplasmic CLU in lung cancer, which might be a potential target for the diagnosis and treatment of metastatic lung cancer.

**Abstract:**

Clusterin (CLU) is a heterodimeric glycoprotein that has been detected in diverse human tissues and implicated in many cellular processes. Accumulating evidence indicates that the expression of secreted CLU correlates with the progression of cancers. However, the molecular mechanisms underlying its tumor-suppressive roles are incompletely uncovered. In this study, we demonstrate that precursor CLU is widely downregulated in lung cancer tissue, in which secretory CLU proteins are slightly decreased. Impressively, overexpressing CLU potently inhibits the migration, invasion and metastasis of lung cancer cells, whereas silencing CLU promotes this behavior; however, it appears that secretory CLU fails to exert similar anti-metastatic effects. Interestingly, the cytoplasmic precursor CLU binds ROCK1 to abrogate the interaction between ROCK1 and ERK and impair ERK activity, leading to the suppression of lung cancer invasiveness. Meanwhile, the expression of CLU was remarkably diminished in lung cancer bone metastasis loci when compared with subcutaneous tumors in the mouse model and hardly detected in the bone metastasis loci of lung cancer patients when compared with the primary. These findings reveal a novel insight into the function and regulation of cytoplasmic CLU in lung cancer, which might be a potential target for the diagnosis and treatment of metastatic lung cancer.

## 1. Introduction

Lung cancer is the leading cause of cancer-related death worldwide [1]. Non-small cell lung cancer (NSCLC) accounts for approximately 85% of all lung cancer cases, which primarily consist of lung adenocarcinoma (LUAD) and lung squamous cell carcinoma (LUSC), while the other 15% of lung cancer cases are small cell lung cancer (SCLC) [2]. Due to atypical symptom expression, nearly half of lung cancer patients show evidence of local or distant metastasis at the time of diagnosis, and the survival of those metastatic patients is only around 8 months [3]. In addition to local metastases in the lymph nodes and contralateral lung, lung cancer cells commonly disperse to diverse organs like bone, brain and liver [4]. Subclonal mutations in patients with LUAD may increase the frequency of postoperative recurrence, implying that it is easier for patients with increased intra-tumor heterogeneity to develop metastases [5]. Despite the fact that chemotherapy is still the main treatment for metastatic lung cancer, the therapeutic effectiveness is usually limited, and chemotherapeutic resistance is another obstacle for cancer therapy. Therefore, it is important to identify novel therapeutic targets and develop effective inhibitors for metastatic lung cancer.

Cytoskeletal reorganization is one of the key steps of tumor cell migration and invasion during tumor metastasis. Previous studies indicate that ROCK1 is generally overexpressed in multiple types of cancers and enhances the migratory ability of tumor cells by actin polymerization [6,7,8,9]. ROCK1 controls cytoskeletal dynamics through phosphorylating and activating LIM kinases (LIMK) and the myosin light chain (MLC), and then phosphorylating actin-depolymerizing protein cofilin [8,10]. Meanwhile, it has been reported that ROCK1 promotes tumor cell migration and invasion through mechanisms independent of cytoskeleton remodeling, such as the upregulation of MMP-9 [11]. Notably, ROCK1 binds and activates ERK to promote the migration of rat vascular smooth muscle cells [12]. A ROCK1 inhibitor (Y-27632) could suppress the TGF-β-induced phosphorylation of ERK [13]. Since the discovery of ROCK1 inhibitors, accumulating evidence supports that ROCK1 could be a potential therapeutic target for cancers, including breast cancer, colon cancer and hepatocellular carcinoma [14,15,16,17]. However, the regulation of the ROCK1 activity in lung cancer and the mechanism through which ROCK1 regulates lung cancer metastasis are not clearly understood.

Clusterin (CLU), a heterodimeric glycoprotein, acts as a molecular chaperone to aid the folding of secreted proteins. Based on CRISPR library screening, CLU is identified as a tumor suppressor in lung cancer [18]. Furthermore, high expression levels of CLU inhibit tumor migration by depressing MMP2 expression [19]. Notably, various isoforms of CLU regulate diverse cellular processes, including apoptosis, cell cycle regulation, DNA repair and resistance against multiple conventional therapies [20,21]. For example, nCLU, a nuclear isoform of CLU, binds Ku70 and triggers apoptosis in the breast cancer cell line MCF-7 by freeing Bax [22]. Contrary to nCLU, intracellular CLU inhibits apoptosis and reduces the sensitivity of chemoresistance in prostate cancer through interacting with Bax specifically [23]. Cytoplasmic precursor CLU (psCLU) is translated from the mRNA, including nine exons. Once cleaved and heavily glycosylated to form the mature isoform, it could be secreted and referred to as secretory CLU (sCLU) [24]. sCLU has been reported to contribute to the resistance of chemotherapeutic agents and has been considered a prognostic biomarker and therapeutic target for various types of cancers [25,26,27,28]. Evidently, biological functions of CLU isoforms are tightly bound to the subcellular localization. It appears that in normal tissues, large amounts of CLU proteins also accumulate in the cytosol. However, whether cytoplasmic CLU exerts biological functions in tumor development and progression remains to be explored.

Here, we demonstrate that cytoplasmic precursor CLU, is widely downregulated in lung cancer tissue. Moreover, overexpressing CLU, especially cytoplasmic CLU potently inhibits the migration, invasion and metastasis of lung cancer cells, whereas silencing CLU promotes these phenomena. At the molecular level, the cytoplasmic precursor CLU binds ROCK1 to abrogate the interaction between ROCK1 and ERK and impair ERK activity, leading to the suppression of lung cancer metastasis. Our findings suggest that cytoplasmic precursor CLU is capable of potently suppressing lung cancer progression by inhibiting the ROCK1/ERK axis, which might provide a potential anti-metastasis strategy for the treatment of metastatic lung cancer.

## 2. Materials and Methods

### 2.1. Tissue Specimens and Immunohistochemistry

Tumor tissues and paired adjacent normal lung tissues were derived from the Cancer Prevention Center of Sun Yat-sen University, and all patients were diagnosed with lung cancer by cytology and pathology in the Department of Pathology. The above specimens were kept in the ultra-low temperature refrigerator at –80 °C since the date of collection. Patient consent and approval were obtained from the Institutional Research Ethics Committee of Sun Yat-sen University prior to the use of these clinical specimens for research. The expression levels of CLU in tissues were detected using immunohistochemistry as previously described [29]. Two independent observers reviewed and scored the degree of immunostaining of formalin-fixed, paraffin-embedded sections according to the positive staining scale and the intensity of staining.

### 2.2. Cell Culture and Recombinant Plasmid Transfection

Human NSCLC cell lines (NCI-H460, NCI-H1299, NCI-H1703 and NCI-H1975) were purchased from the American Type Culture Collection and cultured in DMEM medium (Invitrogen, Carlsbad, CA, USA), supplemented with 10% fetal bovine serum (FBS) (Hyclone, Logan, UT, USA) and 1% penicillin/streptomycin (Invitrogen, Carlsbad, CA, USA). All cell lines were identified by short tandem repeat profiling in the Medicine Laboratory of the Forensic Medicine Department of Sun Yat-sen University. Constructs of pLenti-puro-CLU and pLenti-puro-ROCK1 were produced by inserting the PCR-amplified human CLU and ROCK1 coding sequences into the lentiviral transfection plasmid pLenti. Two CLU-specific shRNA sequences were cloned into the pSuper plasmid to generate pSuper-puro-CLU-sh1 and pSuper-puro-CLU-sh2. Lentivirus preparation and infection experiments have been described in previous reports [30,31]. NSCLC stable cell lines were treated with 0.5 μg/mL puromycin for 23 weeks after 4872 h of infection.

### 2.3. Antibodies and Reagents

Protein expression levels were detected by Western blotting as previously described [32]. Anti-CLU antibody (1:1000; ab92548, Abcam, Cambridge, MA, USA), anti-ROCK1 antibody (1:1000; ab45171, Abcam, Cambridge, MA, USA), anti-Flag antibody (1:2000; F3165, Sigma, Saint Louis, MO, USA), anti-HA antibody (1:2000; H9658, Sigma, Saint Louis, MO, USA), anti-cofilin antibody (1:1000; 5175P, Cell Signaling Technology, Danvers, MA, USA), anti-p-MLC2 (T505) antibody (1:1000; 3671T, Cell Signaling Technology, Danvers, MA, USA), anti-p-LIMK1 (Ser19) antibody (1:1000; 3671T, Cell Signaling Technology, Danvers, MA, USA), anti-LIMK2 antibody (1:1000; 3845, Cell Signaling Technology, Danvers, MA, USA), anti-p-ERK1/2 (T202/Y204) antibody (1:1000; 4370, Cell Signaling Technology, Danvers, MA, USA), anti-ERK1/2 antibody (1:1000; 4695, Cell Signaling Technology, Danvers, MA, USA), anti-GAPDH antibody (1:2000; G8795, Sigma, Saint Louis, MO, USA), Anti-Vinculin antibody (1:10,000; ab129002, Abcam, Cambridge, MA, USA), anti-β-actin antibody (1:2000; A2228, Sigma, Saint Louis, MO, USA) and anti-THOC1 antibody (1:2000; 10920-1-AP, Proteintech, Wuhan, China)were used for Western blotting and immunofluorescence.Y27632 was purchased from MCE (HY-10583, Shanghai, China). SCH772984 was purchased from Selleck (S7101, Shanghai, China).

### 2.4. Cell Migration and Invasion and Transwell Co-Culture Assays

Cell migration and invasion and Transwell co-culture assay were performed using the Transwell system, as previously described [33]. For cell migration and invasion assays, cell suspensions with serum-free DMEM medium were transferred onto the transwell chambers while DMEM medium with 10% FBS was added to the lower chambers. For the Transwell co-culture assay, two NSCLC cell lines, H1299 and H1703, with serum-free DMEM medium, were transferred into the upper chamber, while the indicated cells were deposited into the lower chambers with DMEM medium containing 10% FBS.

### 2.5. Wound-Healing Assay

CLU overexpression or knockdown cell lines were cultured with DMEM medium with 10% FBS in 6-well plates until they reached full confluence to create a confluent monolayer. Subsequently, the cell monolayer was scratched with a (yellow) pipette tip simulating a wound, washed twice with 1X PBS to discard the floating cells and then cultured with serum-free DMEM medium. Initial wounds were imaged immediately after scratching and again every 6 h until the wound healed.

### 2.6. Quantification of CLU by ELISA

CLU overexpression or knockdown cell lines were seeded at the same number as vector control cells in a 6-well plate and cultured for 24 h. The Human CLU ELISA kit (EHCLU, Thermo Fisher Scientific, Waltham, MA, USA) was used to measure CLU protein release in the supernatants of cells according to the manufacturer’s instructions. Absorbance at 450 nm was read on a microplate reader by using a Bio-Tek Synergy 2 microplate reader (Winooski, VT, USA).

### 2.7. Nuclear and Cytoplasmic Protein Extraction Assay

The cytoplasmic and nuclear extracts were isolated by using the NE-PER Nuclear and Cytoplasmic Extraction Reagent Kit (78833, Thermo Fisher Scientific, Waltham, MA, USA) according to the manufacturer’s instructions. The nuclear extract was verified by the expression of P84, which was undetectable in the cytoplasmic extract. CLU expression was detected by using Western blotting analysis to identify the cellular distribution of CLU.

### 2.8. Immunofluorescence and Proximity Ligation Assay (PLA)

Immunofluorescence staining was performed by using an anti-CLU antibody, anti-ROCK1 antibody, Alexa Fluor 555 dye-conjugated secondary antibody and Alexa Fluor 647 dye-conjugated secondary antibody according to the manufacturer’s instructions. The PLA fluorescence assay was carried out using Duolink^®^ PLA reagents (DUO92102, Sigma, Saint Louis, MO, USA) to detect the CLU-ROCK1 interaction in situ (at distances < 40 nm). Immunostaining images were obtained by using a laser scanning microscope (Axioskop 2 plus, Carl Zeiss Co. Ltd., Jena, Germany).

### 2.9. Co-Immunoprecipitation and Mass Spectrometry Analysis

Co-immunoprecipitation (co-IP) was performed based on a standard protocol [34] using Flag magnetic beads (M8823, Sigma, Saint Louis, MO, USA) and HA affinity agarose beads (E6779, Sigma, Saint Louis, MO, USA), and the immunoprecipitans were measured by Western blotting. Mass spectrometry analysis was used to screen CLU-interactive proteins from individual bands after the eluted proteins were separated on SDS–PAGE gels and stained with Coomassie blue.

### 2.10. Animal Models

To investigate the effects of CLU on tumor metastasis, the indicated tumor cells were injected and metastases were monitored by bioluminescent imaging. All animal studies were approved by the SYSU Institutional Animal Care and Use Committee.

### 2.11. Statistical Analysis

All statistical analyses in this article were performed by using the SPSS 20.0 (IBM, Chicago, IL, USA) software package. Comparisons between two groups were performed using Student’s *t* test. *p* values < 0.05 were considered statistically significant, and all error bars in the histograms represent the mean ± SD derived from three independent experiments.

## 3. Results

### 3.1. CLU Downregulation in Lung Cancer Correlates with Poor Patient Survival

To validate the tumor-suppressive role of CLU, we first analyzed the clinical relevance of CLU expression using GEPIA and TCGA lung cancer databases. As shown in Figure 1A, mRNA levels of CLU expression were significantly downregulated in 483 cases of LUAD tissues compared to 347 cases of normal lung tissues, as well as in 486 cases of LUSC tissues compared to 338 cases of normal lung tissues. Similarly, in the TCGA lung cancer database, CLU expression in lung tumor tissues was significantly downregulated compared to those in paired normal lung tissues from 34 LUAD and 49 LUSC patients (Figure 1B).

Moreover, analysis of the K−M plot lung cancer database showed that the overall survival time, as well as the first progression-free survival time of lung cancer patients with high CLU expression, especially LUAD patients, was substantially longer than that of lung cancer patients bearing low CLU expression in their lung tumors (Figure 1C,D). To further validate the protein expression levels of CLU in lung cancer tissues, we performed immunohistochemical staining in 10 pairs of paraffin-embedded lung cancer tissues and adjacent normal lung specimens. The results showed that protein levels of CLU in lung cancer tissues were dramatically lower than those in paracarcinoma lung tissues (Figure 1E,F). Notably, it appeared that in addition to secreted or nuclear CLU, cytoplasmic CLU also accumulated in paracarcinoma lung tissues and decreased in lung cancer tissues. Indeed, immunoblot analysis of eight pairs of lung cancer and adjacent normal lung tissues showed that protein levels of precursor CLU were remarkably decreased in normal lung tissues compared to matched lung tumor tissues (Figure 1G). These data suggest that CLU downregulation in lung cancer correlates with poor patient survival and disease progression.

### 3.2. CLU Especially Cytoplasmic CLU Suppresses Invasion, Migration and Metastasis

To investigate the biological function of CLU in lung cancer, we overexpressed exogenous CLU in two lung cancer cell lines, H1975 and H1299, both of which have almost undetectable CLU protein levels (Figure 2A). Notably, although ELISA assay detected CLU upregulation in the supernatant of CLU-overexpressing cells compared to the vector-control cells (Figure 2B), immunofluorescence assay showed that intracellular CLU was mainly distributed in the cytoplasm (Figure 2C). Consistent with this finding, the cellular component extraction assay showed that intracellular precursor and secretory CLU could both be preferentially localized in the cytosol (Figure 2D). Compared to the vector-control cells, H1975 and H1299 cells overexpressing CLU had compromised migration and invasion abilities, as evaluated by Transwell assays (Figure 2E). When injected ventricularly into nude mice, the vector-control H1299 cells displayed obviously metastatic signals in various body sites, whereas CLU-overexpressing H1299 cells were hardly able to develop metastatic loci (Figure 2F). H&E staining confirmed the tumor metastases of the vector-control H1299 cells, but not the CLU-overexpressing H1299 cells in tissues of the lung, cervical lymph node and femur (Figure 2G). These data demonstrate the potent anti-metastatic role of CLU in lung cancer. To further explore whether extracellular secretory CLU or intracellular CLU has this anti-metastatic function, we established stable cell lines overexpressing CLU depleted of a signal peptide (H1299-CLU-del SP and H1975-CLU-del SP). CLU depleted of the signal peptide indeed intracellularly accumulated and still markedly suppressed the migration and invasion of lung cancer cells (Figure 2H,I). We designed another assay to detect whether the secretory or intracellular CLU repressed migration using supernatant culture medium from the vector-control or CLU-overexpressing cells (Figure 2J). As shown in Figure 2K, the supernatant culture medium from the vector-control or CLU-overexpressing cells hardly altered the migratory ability of lung cancer cells. These results suggest that cytoplasmic CLU exerts the potent anti-metastatic effects in lung cancer.

### 3.3. Silencing CLU Promotes Lung Cancer Invasion and Migration

Silencing CLU with distinct shRNAs in H460 cells dramatically reduced the protein levels of intracellular CLU, whereas interestingly, secretory CLU was slightly downregulated in CLU-silenced H460 cells compared to vector-control H460 cells (Figure 3A,B). Immunofluorescence staining further demonstrated apparent intracellular CLU intensity in the vector-control H460 cells, while intracellular CLU was basically diminished in CLU-silenced H460 cells (Figure 2C). Moreover, CLU knockdown in H460 cells markedly increased migration and invasion abilities (Figure 2D). In parallel, a wound-healing assay further showed that the wounds of the vector-control H460 cells healed much slower than those of the CLU-silenced H460 cells (Figure 2E), consistently supporting the pro-migratory role of silencing CLU in lung cancer cells.

### 3.4. CLU Interacts with ROCK1 to Suppress Metastatic Traits

To explore the mechanism of the cytoplasmic CLU-mediated inhibition of lung cancer metastasis, immunoprecipitation–mass spectrometry (IP–MS) analysis of proteins interacting with cytoplasmic CLU was conducted. We found that a metastasis-related regulator ROCK1 emerged as one of the CLU-interacting proteins (Figure 4A,B). The co-immunoprecipitation assay in 293FT cells transiently overexpressing Flag-tagged CLU and HA-tagged ROCK1 validated the mutual interaction between precursor CLU and ROCK1, whereas it appeared that secretory CLU was unable to bind ROCK1(Figure 4C). Immunofluorescent co-staining of both CLU and ROCK1 in CLU-overexpressing cells showed that CLU co-localized with ROCK1 in the cytosol (Figure 4D). Proximity ligation assay (PLA) also detected the mutual interaction between CLU and ROCK1 in the cytoplasm of lung cancer cells (Figure 4E), indicating that cytoplasmic CLU might function by regulating ROCK1.

To validate this notion, we employed a traditional inhibitor of ROCK1, namely Y27632, in CLU-silenced lung cancer cells. As shown in Figure 4F,G, inhibiting ROCK1 through Y27632 treatment significantly abrogated the pro-invasive and pro-migratory effects of silencing CLU in lung cancer cells. As a key effector that regulates stress fibers and focal adhesions, ROCK1 phosphorylation activates LIMK1/2, which, in turn, phosphorylates cofilin and suppresses its actin-depolymerizing activity, thereby stabilizing actin stress fibers. Notably, neither overexpressing CLU nor silencing CLU exerted an impact on cellular morphology or the cytoskeleton (Figure 4H). Moreover, neither overexpressing CLU nor silencing CLU changed the ROCK1 protein levels or impacted the conventional downstream pathways of ROCK1, as evidenced by the activity or expression levels of myosin light chain 2 (MLC2), LIMK1/2 and coffin (Figure 4I). These data indicate that CLU interacts with ROCK1 to suppress metastatic traits without affecting the conventional downstream pathways of ROCK1.

### 3.5. CLU Inhibits Metastasis through the ROCK1/ERK Axis

Previous data reported that ROCK1 might directly phosphorylate ERK1/2 to enhance ERK1/2 activity in an unconditional manner. Interestingly, overexpression of CLU in H1299 and H1975 cells markedly inhibited the phosphorylation of ERK1/2, whereas knockdown of CLU in H460 cells dramatically promoted ERK1/2 phosphorylation (Figure 5A). Moreover, inhibiting ROCK1 through treatment with its inhibitor Y27632 remarkably diminished the CLU knockdown-induced upregulation of ERK1/2 phosphorylation in H460 cells (Figure 5B). Then we performed a co-immunoprecipitation assay and found that ROCK1 could bind to ERK1/2 in the absence of CLU and CLU almost completely abrogated the interaction between ROCK1 and ERK1/2 (Figure 5C), indicating that CLU might bind ROCK1 to competitively inhibit ROCK1-mediated phosphorylation of ERK1/2. Furthermore, an ERK inhibitor, SCH772984, which was used to determine whether the ERK1/2 activity significantly contributed to the pro-metastatic effects of CLU silencing in lung cancer cells, dramatically inhibited ERK1/2 phosphorylation and suppressed the migration and invasion of CLU-silenced lung cancer cells (Figure 5D–F). These data indicate that CLU inhibits lung cancer progression through the ROCK1/ERK axis.

### 3.6. CLU Is Negatively Related to Metastasis in Models and Tissues

To further validate the potent anti-metastatic role of cytoplasmic CLU, we measured the expression levels of CLU in subcutaneous tumors and bone metastasis loci by immunohistochemical staining (Figure 6A–C). CLU was significantly suppressed in bone metastasis loci compared to the subcutaneous tumors in vivo.

In addition, we compared the expression levels of CLU in parental A549 and PC9 cells with their matched bone-metastatic cancer cells, primarily cultured from their corresponding bone-specific metastases (BM). The expression of CLU was dramatically diminished in bone-metastatic cancer cells compared to their parental A549 and PC9 cells (Figure 6D). Moreover, the overexpression of CLU in bone-metastatic cancer cells not only apparently inhibited the invasion and migration of these bone-metastatic cancer cells, but also decreased the phosphorylation of ERK1/2 without affecting ROCK1 expression levels (Figure 6E,F). Importantly, in the case of metastatic lung cancer patients, CLU expression could be detected in primary lung tumor tissue but was hardly detected in the bone metastasis loci (Figure 6G). Collectively, these results suggest that CLU is able to bind ROCK1 to block the ROCK1/ERK axis and inactivates ERK1/2 activity, leading to the potent suppression of migration, invasion and metastasis in lung cancer (Figure 6H).

## 4. Discussion

In this paper, we determine that CLU, especially its precursor form that is downregulated in lung cancer, functions as a tumor suppressor by inhibiting the migration, invasion and metastasis of lung cancer cells. Interestingly, instead of secretory CLU, cytoplasmic CLU exerts a potent anti-metastatic effect on lung cancer cells. Mechanistically, cytoplasmic precursor CLU binds ROCK1 to decrease the phosphorylation of ERK1/2 by inhibiting the kinase activity of ROCK1, leading to an anti-metastatic effect in lung cancer cells.

Diverse functions of CLU have been reported in a number of physiologic and pathologic processes, including cancer [35]. Our work demonstrates an anti-metastatic role for CLU through a new pathway in lung cancer. It has been further reported that CLU inhibits lung cancer progression by inhibiting TGFBR1-induced TRAF6/TAB2/TAK1 complex recruitment and thus blocking the TAK1-NF-κB axis [36], indicating its tumor-suppressive function, which is consistent with our in vivo data. Furthermore, Olesya et al. reported that mice with their CLU gene disrupted were more prone to developing neuroblastomas than normal control mice [37]. In our current study, it is noteworthy that the downregulation of CLU is closely associated with tumor metastasis, according to the analysis of TCGA lung cancer datasets, as well as the anti-metastatic effect of CLU overexpression both in vitro and in vivo. It has been recognized that more than half of lung cancer patients show evidence of local or distant metastasis at the time of diagnosis, and the median survival of these patients is only approximately 8 months [3]. Our results show that CLU is commonly downregulated in lung cancer and its downregulation correlates with poor patient survival and disease progression, indicating that CLU downregulation significantly contributes to lung cancer development and progression and might represent an early diagnostic marker for metastatic lung cancer patients.

Of note, the CLU gene produces different isoforms of CLU, which show distinct patterns of subcellular localization and diverse biological functions [22,38,39]. For example, nuclear CLU (nCLU) reduces cell migration by binding to α-actinin. a cytoskeletal actin-binding protein, in prostate cancer cells [40]. Similarly, we find that cytoplasmic CLU regulates cancer metastasis through an unconventional pathway. Maurizio et al. showed that intracellular CLU induces cell cycle arrest and cell death by inhibiting the cyclin B1/CDK1 complex in prostate cancer cells [39]. In this context, we demonstrate that rather than secretory or nuclear CLU, cytoplasmic CLU is significantly decreased in lung cancer cells, which potently suppresses migration, invasion and metastasis; this suggests that cytoplasmic CLU is the primary functional isoform exerting anti-metastatic effects in lung cancer.

Previous studies have shown that ROCK1 regulates cancer cell motility, invasion, migration and metastasis [41,42,43]. Moreover, our current study reveals that cytoplasmic precursor CLU binds ROCK1 to abrogate the interaction between ROCK1 and ERK, leading to the suppression of lung cancer migration and invasion. Consistent with previous reports showing that the treatment of the ROCK1 inhibitor Y27632 markedly inhibits the invasion and migration of breast cancer cells in vitro and bone metastases in vivo [15], our study demonstrates that Y27632 also inhibits the migration of lung cancer cells, especially those silenced with CLU expression. As a kinase, ROCK1 significantly induces actin polymerization and increases migratory ability through phosphorylating and activating LIMK and MLC and phosphorylating actin-depolymerizing protein cofilin, consequently leading to cytoskeletal reorganization [10]. However, our study finds that the binding of CLU to ROCK1 fails to affect the activity or expression of cytoskeleton-reorganizing proteins, such as LIMK and cofilin. Notably, it is also reported that ROCK1 can phosphorylate ERK2 to promote tumor metastasis [12,44]. Consistent with this report [14], we demonstrate that the binding of CLU to ROCK1 abrogates the interaction between ROCK1 and ERK and thus, inhibits ERK activation. Moreover, the inhibition of ROCK1 through treatment with its inhibitor Y27632 remarkably diminishes the upregulation of ERK1/2 phosphorylation induced by CLU knockdown.

## 5. Conclusions

Cytoplasmic CLU is downregulated in lung cancer and correlates with poor survival. Cytoplasmic precursor CLU inhibits lung cancer metastasis by binding ROCK1 to decrease the phosphorylation of ERK1/2. This work reveals a novel function and regulation of the cytoplasmic precursor CLU in lung cancer, which might be a potential target for the diagnosis and treatment of metastatic lung cancer. Furthermore, our findings extend the current understanding of the kinase activity of ROCK1 without affecting cellular cytoskeleton and identify a novel mechanism underlying ROCK1-mediated ERK phosphorylation and activation due to CLU loss during lung cancer progression.

## Figures and Tables

**Figure 1 cancers-14-02463-f001:**
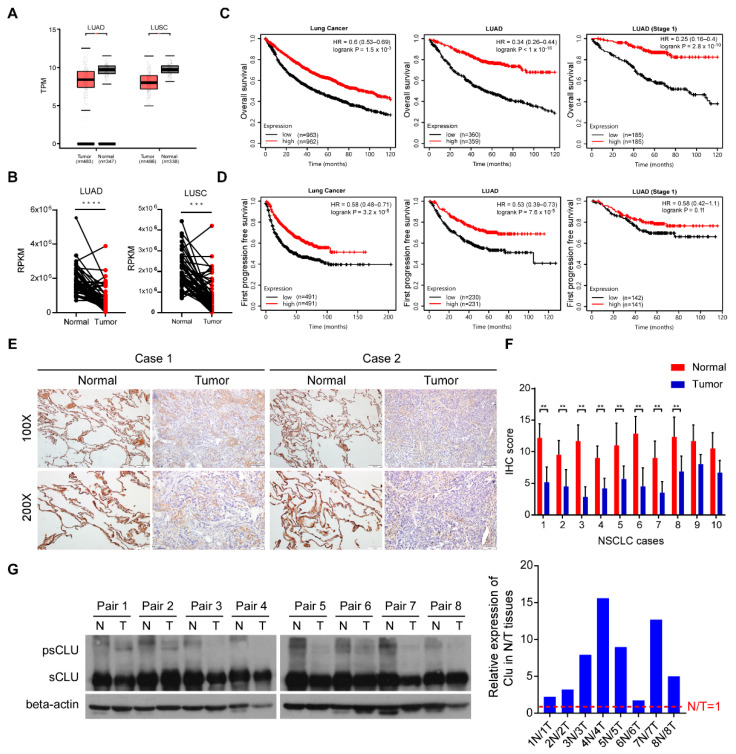
CLU downregulation in lung cancer correlates with poor patient survival. (**A**) The expression of CLU in GEPIA (Gene Expression Profiling Interactive Analysis) datasets. (**B**) The expression of CLU in 34 pairs of LUAD tumor tissues and 49 LUSC tumor tissues versus matched non-tumorous lung tissues in the K−M plot lung cancer database. (**C**) Kaplan-Meier analysis of overall survival in TCGA lung cancer, LUAD and LUAD (stage 1) cohort based on the expression of CLU. (**D**) Kaplan-Meier analysis of first progression-free survival in TCGA lung cancer, LUAD and LUAD (stage 1) cohort based on the expression of CLU. The divergence of patient survival between CLU high- and low-expression patients was statistically significant within the above cancer types. (**E**) Typical photographs of IHC staining on CLU expression in the lung cancer and adjacent normal tissues. (**F**) IHC score of CLU expression in lung cancer and paired adjacent normal tissues (n = 10). (**G**) CLU protein levels in 8 pairs of lung cancer and adjacent normal tissue proteins, and β-actin was used as a loading control. The band at 68 kDa indicated the presence of precursor CLU (psCLU) and the band at 37 kDa indicated the presence of secretory CLU (sCLU). Results are presented as mean ± SD, ** *p* < 0.01, *** *p* < 0.001, **** *p* < 0.0001.

**Figure 2 cancers-14-02463-f002:**
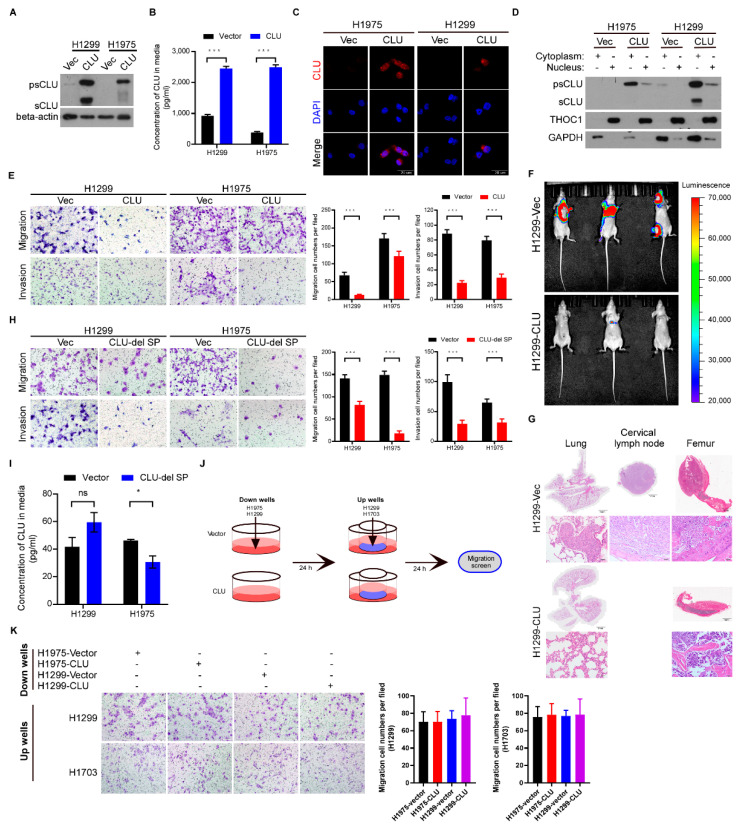
CLU, especially cytoplasmic CLU, suppresses invasion, migration and metastasis. (**A**) The protein levels of psCLU and sCLU in H1975 and H1299 vectors (vec) and CLU-overexpressing cells. GAPDH was used as a loading control. (**B**) ELISA assay shows the concentration of sCLU in H1975 and H1299 vec and CLU-overexpressing cells. (**C**) Representative micrographs of the staining of intracellular CLU in H1975 and H1299 cells. (**D**) WB analysis of subcellular distribution of psCLU and sCLU in H1975 and H1299 cells. (**E**) Representative images of Transwell assay and relative quantification of the migration or invasion cells in H1299 and H1975 cells. (**F**,**G**) H1299-Vec or H1299-CLU cells were injected via cardiac ventricle into nude mice (n = 5 per group). Representative bioluminescent images of systemic metastasis and ex vivo organ metastases are shown (**F**), H&E histologically confirmed tumor cells in bone and brain tissue (**G**). (**H**) Representative images of Transwell assay and relative quantification of the migration or invasion cells in H1975-Vec, H1975-CLU-del SP cells, H1299-Vec and H1299-CLU-del SP cells. (**I**) ELISA assay shows the concentration of CLU in indicated cells. (**J**) Vector-control or CLU-overexpressing cells were seeded in the down wells and cultured with the supernatant medium of H1703 or H1299 cells in the up wells. (**K**) Representative images of Transwell assay and relative quantification of migration cells in H1299 and H1703 cells stimulated by indicated supernatant samples. Results are presented as mean ± SD, * *p* < 0.05, *** *p* < 0.001, ns means no significance.

**Figure 3 cancers-14-02463-f003:**
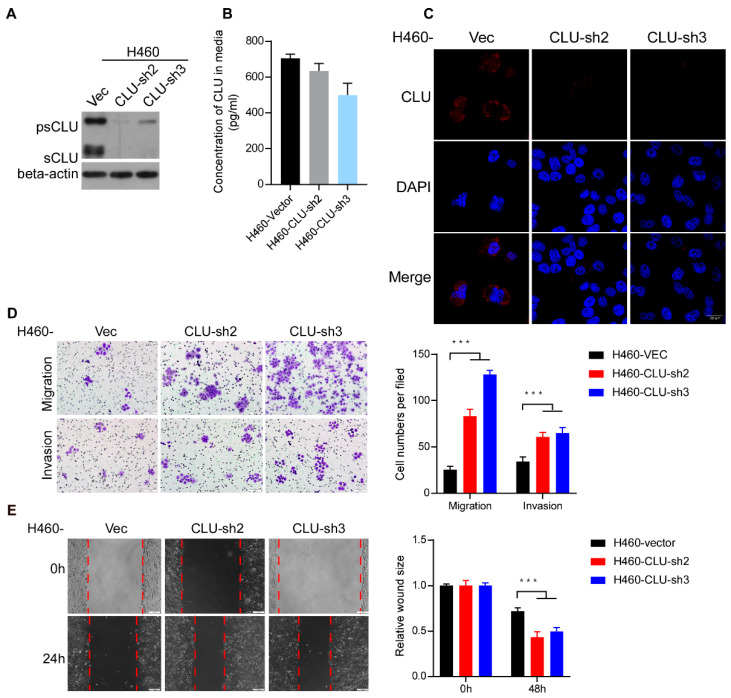
Silencing CLU promotes lung cancer invasion and migration. (**A**) The protein level of CLU in CLU-silenced H460 cells compared to vector-control cells. GAPDH was used as a loading control. (**B**) The concentration of sCLU in H460 cells was detected by ELISA assay. (**C**) Representative micrographs of immunofluorescence for intracellular CLU distribution in CLU-silenced H460 cells and vector-control cells. (**D**) Representative images of Transwell assay and relative quantification of the migration or invasion cells in CLU-silenced H460 cells and vector-control cells. (**E**) Representative images of wound-healing assay and relative wound size in CLU-silenced H460 cells and vector-control cells. Results are presented as mean ± SD, *** *p* < 0.001, ns means no significance.

**Figure 4 cancers-14-02463-f004:**
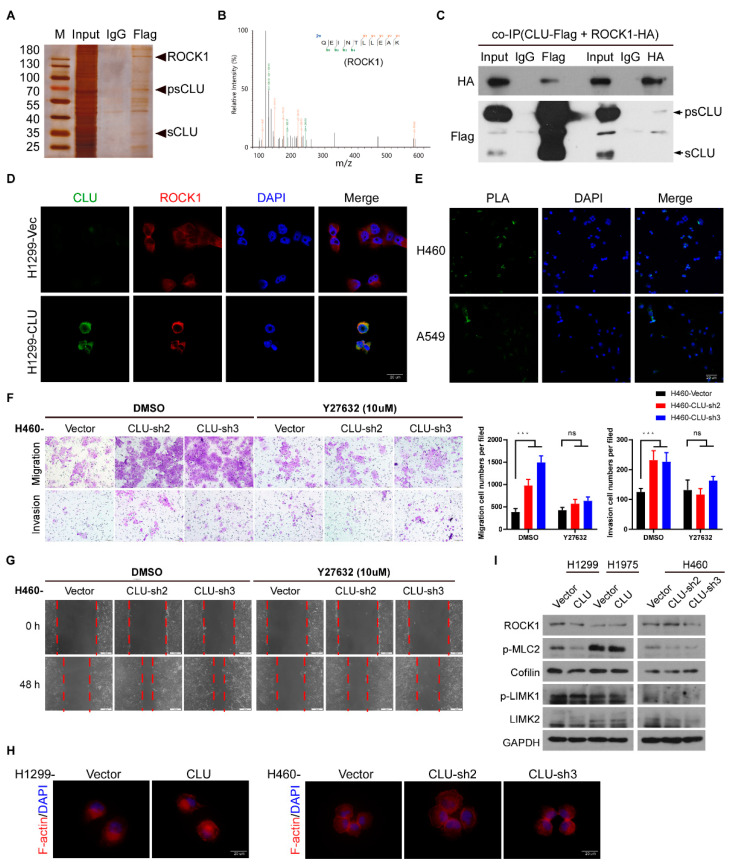
CLU interacts with ROCK1 to suppress metastatic traits. (**A**,**B**) IP–MS analysis was performed using 293FT cells with ectopic expression of CLU-Flag. Metastasis-related regulator ROCK1 was identified as a CLU binding protein. (**C**) Co-IP assay was performed in 293FT cells to confirm the interaction between precursor CLU and ROCK1. (**D**) Representative images of immunofluorescent co-staining of intracellular CLU and ROCK1 in CLU-overexpressing and vector-control cells. (**E**) PLA assay was performed to confirm the interaction of CLU and ROCK1 in H460 and A549 cells. (**F**,**G**) The effect of Y27632 treatment in CLU-silenced H460 cells and vector-control cells on their migration and invasion abilities. (**H**) Cell morphology had no difference in CLU-overexpressing H1299 cells and CLU-silenced H460 cells. F-actin was used to detect the cytoskeleton in those cells. (**I**) The effect of CLU on ROCK1, p-MLC2, Cofilin, p-LIMK1 and LIMK2 in CLU-overexpressing H1299 and H1975 cells and CLU-silenced H460 cells. GAPDH was used as a loading control. Results are presented as mean ± SD, *** *p* < 0.001, ns means no significance.

**Figure 5 cancers-14-02463-f005:**
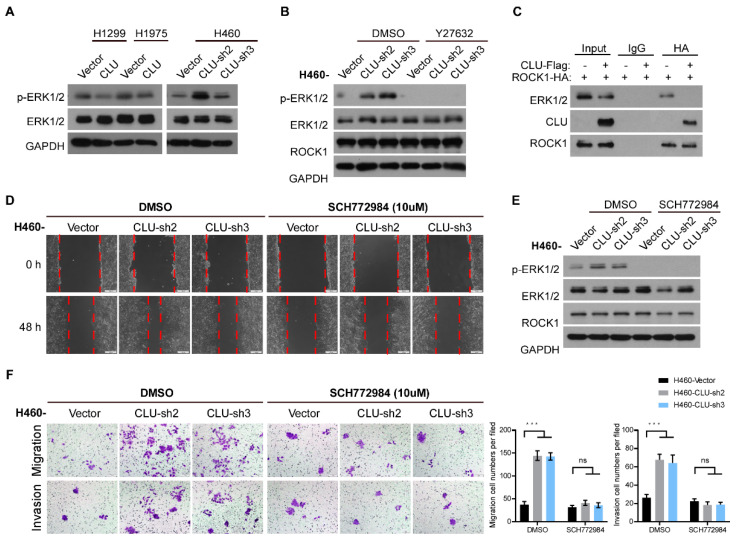
CLU inhibits metastasis through the ROCK1/ERK axis. (**A**) The effect of overexpressing or silencing CLU on the level of phosphorylation of ERK in indicated cells. (**B**) The effect of Y27632 treatment on the level of phosphorylation of ERK in CLU-silenced H460 cells and vector-control cells. (**C**) Co-IP assay was performed to evaluate the interaction between CLU, ROCK1 and ERK1/2. D–F. The effect of SCH772984 (ERK inhibitor) in CLU-silenced H460 cells and vector-control cells on their migration and invasion abilities (**D**,**F**) and the level of phosphorylation of ERK (**E**). Results are presented as mean ± SD, *** *p* < 0.001, ns means no significance.

**Figure 6 cancers-14-02463-f006:**
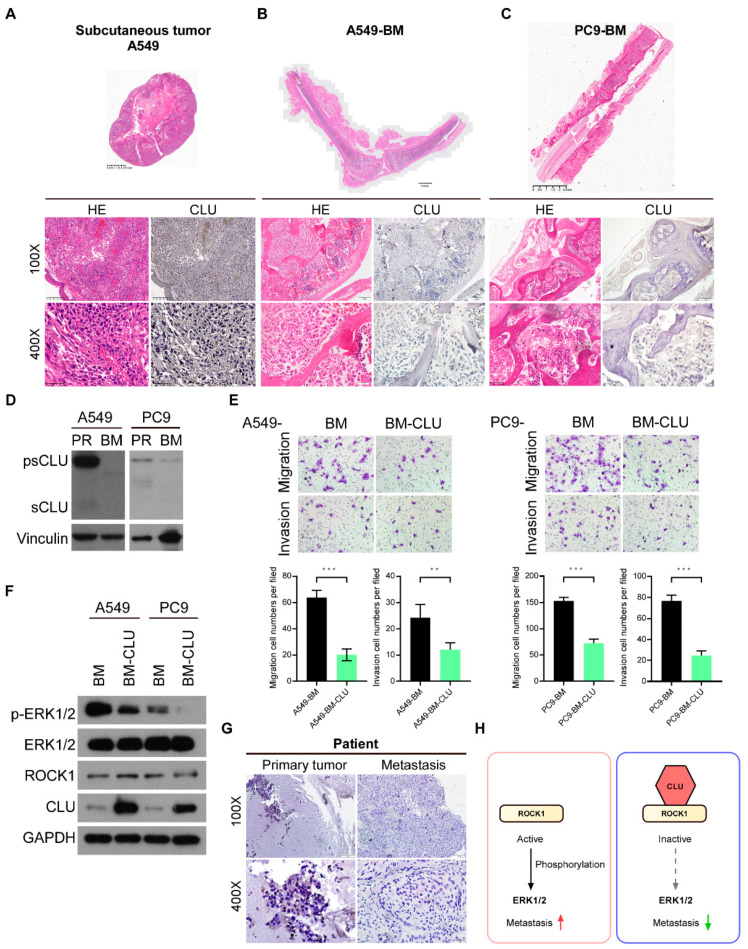
Precursor CLU is negatively related to metastasis in models and tumor tissues. (**A**–**C**) Representative images of H&E and IHC staining for CLU subcutaneous tumor lesions of A549 cells (**A**), bone metastases loci of A549-BM cells (**B**) and PC9-BM cells (**C**). (**D**) The expression of CLU in A549- and PC9-BM cells and parental (PR) cells. (**E**). The effect of silenced CLU in A549- and PC9-BM cells on their abilities of migration and invasion. (**F**) The effect of silenced CLU on the level of phosphorylation of ERK in A549- and PC9-BM cells. (**G**) Representative images of IHC staining for CLU in lung cancer primary and bone metastatic loci. (**H**) The mechanism through which CLU inhibits metastasis is via the ROCK1/ERK axis. Results are presented as mean ± SD, ** *p* < 0.01, *** *p* < 0.001, ns means no significance.

## Data Availability

Not applicable.

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
