# Peer review of "Cytoplasmic Clusterin Suppresses Lung Cancer Metastasis by Inhibiting the ROCK1-ERK Axis"

_cancers, 2022, doi:10.3390/cancers14102463_

Round 1

Reviewer 1 Report

The manuscript "Cytoplasmic Clusterin suppresses lung cancer metastasis by inhibiting ROCK1-ERK axis" and contains new data on the role of cytoplasmic clusterin in metastasis.

Minor

Why are cell lines purchased from ATCC not cultured in the media recommended by the supplier?

Author Response

Dear Reviewers:

We are very grateful to Reviewers for reviewing the paper so carefully. We have carefully considered the suggestion of Reviewers and make some changes in revised manuscript. In the following section, we summarize our responses to each comment from the reviewers. We believe that our responses have well addressed all your concerns from.

Response to Reviewer 1 Comments

Point: Why are cell lines purchased from ATCC not cultured in the media recommended by the supplier?

Response: We thank the reviewer for raising these issues. Indeed, we have cultured these cell lines which purchased from ATCC with RPMI-1640, not only with DMEM medium, respectively, and found the status of tumor cells were unaffected. As the same time, DMEM medium has been widely used in human NSCLC cell lines NCI-H460, NCI-H1299, NCI-H1703 and NCI-H1975 cells (Nat Commun. 2021;12(1):2693; Acta Pharm Sin B. 2019;9(3):516-525; J Exp Clin Cancer Res. 2019;38(1):333.). In addition, due to the influence of COV-19, we lost stabilized supplements of RPMI-1640. Therefore, these cell lines purchased from ATCC in this work were not cultured in RPMI-1640, the media recommended by the supplier.

Reviewer 2 Report

Major and specific comments

In this study, authors aimed to explore the role of clusterin in suppressing lung cancer metastasis. Their findings showed that expression of clusterin was downregulated in tumor tissue and was associated overall survival, and overexpression of clusterin inhibited the metastatic potential of lung cancer cells via interaction with ROCK. Generally, this study is well designed, the experiments have been properly conducted, and the conclusions can be supported by the results. These findings clearly extend our knowledge to understanding the mechanistic role of clusterin and its localization in anti-metastasis of lung cancer cell. In addition, the description and wording are concise and fluent.

Author Response

Dear Reviewers:

We are very grateful to Reviewers for reviewing the paper so carefully. We have carefully considered the suggestion of Reviewers and make some changes in revised manuscript. In the following section, we summarize our responses to each comment from the reviewers. We believe that our responses have well addressed all your concerns from.

Response to Reviewer 2 Comments

Point: In this study, authors aimed to explore the role of clusterin in suppressing lung cancer metastasis. Their findings showed that expression of clusterin was downregulated in tumor tissue and was associated overall survival, and overexpression of clusterin inhibited the metastatic potential of lung cancer cells via interaction with ROCK. Generally, this study is well designed, the experiments have been properly conducted, and the conclusions can be supported by the results. These findings clearly extend our knowledge to understanding the mechanistic role of clusterin and its localization in anti-metastasis of lung cancer cell. In addition, the description and wording are concise and fluent.

Response: We fell great thanks to you for reading our paper carefully and giving the above positive comments.

Reviewer 3 Report

Comments

The manuscript is an interesting study about the evaluation of the correlation between expression of clusterin and lung cancer progression, thus providing insight about potential target for diagnosis and treatment of metastatic lung cancer.

The topic of the paper is worthy of investigation and well fits with the aim of Cancers. Before further processing of the paper, authors are asked to take into consideration the following issues, listed as per article section.

Abstract

In its present form it is too literal, please consider the possibility to revise the section by adding some key data results.

Introduction

Here, authors presented lung cancer (lines 36-50), Clusterin (lines 51-69) and ROCK1 and ROCK2 (lines 70-80). This reviewer totally agree with authors that it is important to focus the main “actor” of the study, but is wondering if authors could improve the section by highlighting the novelty of their approach to strengthen the impact of their study in the scientific community.

Materials and Methods

No specific comments to this section

Results

Authors state: we first analyzed the clinical relevance of CLU expression in lung cancer tissue using various lung cancer databases (lines 183-184). Can further information be added here?

Discussion

Authors should insert some consideration about the importance of their finding compared with the available knowledge in the field.

Conclusion

Can authors improve the section by highlighting the key outcome of the study and the potential impact among the scientific community?

Author Response

Dear Reviewers:

We are very grateful to Reviewers for reviewing the paper so carefully. We have carefully considered the suggestion of Reviewers and make some changes in revised manuscript. In the following section, we summarize our responses to each comment from the reviewers. We believe that our responses have well addressed all your concerns from.

Response to Reviewer 3 Comments

Comments

The manuscript is an interesting study about the evaluation of the correlation between expression of clusterin and lung cancer progression, thus providing insight about potential target for diagnosis and treatment of metastatic lung cancer.

The topic of the paper is worthy of investigation and well fits with the aim of Cancers. Before further processing of the paper, authors are asked to take into consideration the following issues, listed as per article section.

Point 1: Abstract

In its present form it is too literal, please consider the possibility to revise the section by adding some key data results.

Response 1: Thank you for pointing out this problem in our manuscript. Corresponding description have been incorporated into the revised manuscript as follow:

Abstract: Clusterin (CLU) is a heterodimeric glycoprotein that has been detected in diverse human tissues and implicated in many cellular processes. Accumulating evidence indicates that the expression of secreted CLU correlates with the progression of cancers. However, the molecular mechanisms underlying its tumor-suppressive roles are incompletely uncovered. In this study, we demonstrate that both mRNA and protein levels of (delete these words in green). precursor CLU are widely downregulated in lung cancer tissue, in which secretory CLU proteins are slightly decreased. Impressively, overexpressing CLU potently in-hibits, whereas silencing CLU promotes migration, invasion and metastasis of lung cancer cells, whereas it appears that secretory CLU fails to exert similar anti-metastatic effects. Interestingly, the cytoplasmic precursor CLU binds ROCK1 to abrogate the interaction between ROCK1 and ERK and to impair ERK activity, leading to suppression of lung cancer invasiveness. Meanwhile, the expression of CLU was remarkably diminished in lung cancer bone metastasis loci as com-pared to the subcutaneous tumors in mouse model, and hardly detected in the bone metastasis loci of lung cancer patients as compared to the primary (add this sentence in blue text). These findings reveal a novel insight into the function and regulation of cytoplasmic CLU in lung cancer, which might be a potential target for diagnosis and treatment of metastatic lung cancer.

Point 2: Introduction

Here, authors presented lung cancer (lines 36-50), Clusterin (lines 51-69) and ROCK1 and ROCK2 (lines 70-80). This reviewer totally agree with authors that it is important to focus the main “actor” of the study, but is wondering if authors could improve the section by highlighting the novelty of their approach to strengthen the impact of their study in the scientific community.

Response 2: We thank the reviewer for pointing out this issue. To address the reviewer’s concern, in this revision, we have improved the structure and organization of the manuscript. We have deleted some redundant descriptions, and made supplementary explanations for some confusing descriptions in the section of introduction. The revised introduction focus on lung cancer metastasis (lines 36-54), Cytoskeletal reorganization and ROCK1 (lines 55-69), Clusterin (lines 70-87). The details are marked as green words in the following text.

Introduction

Lung cancer is the leading cause of cancer-related death worldwide [1]. Non-small cell lung cancer (NSCLC) accounts for approximately 85% of all lung cancer cases, which primarily consist of lung adenocarcinoma (LUAD) and lung squamous cell carcinoma (LUSC), while the other 15% of lung cancer cases are small cell lung cancer (SCLC) [2]. Because of the atypical symptoms, nearly half of lung cancer patients show evidence of local or distant metastasis at the time of diagnosis, and the survival of those metastatic patients is only around 8 months [3]. In addition to local metastasis in the lymph nodes and contralateral lung, lung cancer cells commonly disperse to diverse organs like bone, brain and liver [4]. Subclonal mutations in patients with LUAD may increase the frequency of postoperative recurrence, implying that it is easier for patients with increased intra-tumor heterogeneity to develop metastasis in the early stage [5]. Despite that chemotherapy still is the main treatment for metastatic lung cancer, the therapeutic effectiveness is usually limited and chemotherapeutic resistance is another obstacle for cancer therapy. Therefore, it is important to identify novel therapeutic targets for metastatic lung cancer and develop effective inhibitors.

Cytoskeletal reorganization is one of the key steps of tumor cell migration and invasion during tumor metastasis. Previous studies indicate that ROCK1 is generally overexpressed in multiple types of cancers and enhances the migratory ability of tumor cells by actin polymerization [6-9]. ROCK1 controls of cytoskeletal dynamics through phosphorylating and activating LIM kinases (LIMK) and myosin light chain (MLC), followed by phosphorylating actin-depolymerizing protein cofilin [8,10]. Meanwhile, it has been reported that ROCK1 promotes tumor cells migration and invasion through the mechanism independent of cytoskeleton remodeling, such as the upregulation of MMP-9 [11]. Noteworthily, ROCK1 binds and activates ERK to promote the migration of rat vascular smooth muscle cells [12]. ROCK1 inhibitor (Y-27632) could suppress TGF-β-induced phosphorylation of ERK [13]. Since the discovery of ROCK1 inhibitors, accumulating evidence supports that ROCK1 could be a potential therapeutic target for cancers, including breast cancer, colon cancer and hepatocellular carcinoma [14-17]. However, the regulation of the ROCK1 activity in lung cancer and the mechanism of ROCK1 regulating lung cancer metastasis are not clearly understood.

Clusterin (CLU), a heterodimeric glycoprotein, acts as a molecular chaperone to aid the folding of secreted proteins. Based on CRISPR library screening, CLU is identified as a tumor suppressor in lung cancer [18]. Furthermore, high expression levels of CLU inhibit tumor migration by depressing MMP2 expression [19]. Notably, various isoforms of CLU regulate diverse cellular processes, including apoptosis, cell cycle regulation, DNA repair and resistance against multiple conventional therapies [20,21]. For example, nCLU, a nu-clear isoform of CLU, binds Ku70 and triggers apoptosis in the breast cancer cell line MCF-7 by freeing Bax [22]. Contrary to nCLU, intracellular CLU inhibits apoptosis and reduces the sensitivity of chemoresistance in prostate cancer through specifically inter-acting with Bax [23]. Cytoplasmic precursor CLU (psCLU) is translated from the mRNA, including nine exons. Once cleaved and heavily glycosylated to form the mature isoform, it could been secreted, and referred to as secretory CLU (sCLU) [24]. sCLU has been reported to contribute to the resistance of chemotherapeutic agents, and considered as a prognostic biomarker and therapeutic target in various types of cancers [25-28]. Evidently, biological functions of CLU isoforms are tightly bound to the subcellular localization. It appears that in normal tissues, large amounts of CLU proteins also accumulate in the cytosol. However, whether cytoplasmic CLU exerts biological functions in tumor development and progression remains to be explored.

Here, we demonstrate that cytoplasmic precursor CLU, is widely downregulated in lung cancer tissue. Moreover, overexpressing CLU, especially cytoplasmic CLU potently inhibits, whereas silencing CLU promotes migration, invasion and metastasis of lung cancer cells. At the molecular level, the cytoplasmic precursor CLU binds ROCK1 to abrogate the interaction between ROCK1 and ERK and to impair ERK activity, leading to suppression of lung cancer metastasis. Our findings suggest that cytoplasmic precursor CLU is capable of potently suppressing lung cancer progression by inhibiting ROCK1/ERK axis, which might provide a potential anti-metastasis strategy for the treatment of metastatic lung cancer.

Point 3: Results

Authors state: we first analyzed the clinical relevance of CLU expression in lung cancer tissue using various lung cancer databases (lines 183-184). Can further information be added here?

Response 3: Thank you for your comment. We have made some for detail information in the revised manuscript as follow.

“To validate the tumor-suppressive role of CLU, we first analyzed the clinical relevance of CLU expression in lung cancer tissue using various lung cancer databases.” change to “To validate the tumor-suppressive role of CLU, we first analyzed the clinical relevance of CLU expression using GEPIA and TCGA lung cancer databases.” (further informations of the databases are described in the figure legends)

Point 4: Discussion

Authors should insert some consideration about the importance of their finding compared with the available knowledge in the field.

Response 4: Thank you for the above suggestion. In the revised manuscript, we have inserted some consideration about the importance of our finding in section 2, 3 and 4 of discussion.

Section 2 (lines 391-407)

Diverse functions of CLU have been reported in a number of physiologic and pathologic processes, including cancer [35]. Our work demonstrates an anti-metastatic effect role of CLU through a new pathway in lung cancer. It has been further reported that CLU inhibits lung cancer progression via inhibiting TGFBR1-induced TRAF6/TAB2/TAK1 complex recruitment and thus blocking the TAK1-NF-κB axis [36], indicating its tumor-suppressive function and consistent with our in vivo data. (we add these sentences to compare our work with others.) Furthermore, Olesya et al. reported that mice with CLU gene disrupted were more prone to developing neuroblastoma than control normal mice [37]. In our current study, it is noteworthy that downregulation of CLU is closely associated with tumor metastasis, according to the analysis of TCGA lung cancer datasets, as well as the anti-metastatic effect of CLU overexpression both in vitro and in vivo. It has been recognized that more than half of lung cancer patients show evidence of local or distant metastasis at the time of diagnosis, and the median survival of these patients is only approximately 8 months [3]. Our results show that CLU is commonly downregulated in lung cancer and its downregulation correlates with poor patient survival and disease progression, indicating that CLU downregulation importantly contributes to lung cancer development and progression and might represent an early diagnostic marker for metastatic lung cancer patients.

Section 3 (lines 408-418)

Of note, CLU gene produce different isoforms of CLU, which show distinct patterns of subcellular localization and pose diverse biological functions [22, 38, 39]. For example, nuclear CLU (nCLU) reduces cell migration through binding to α-actinin. a cytoskeletal actin-binding protein, in prostate cancer cells [40]. Similarly, we find that cytoplasmic CLU regulate cancer metastasis in an unconventional pathway. (insert the comparison). Maurizio et al. showed that intracellular CLU induces cell cycle arrest and cell death via inhibiting Cyclin B1/CDK1 complex in prostate cancer cells [39]. In this context, we demonstrate that other than secretory or nuclear CLU, cytoplasmic CLU is significantly decreased in lung cancer cells, which potently suppresses migration, invasion and metastasis suggesting that cytoplasmic CLU is the primary functional isoform exerting anti-metastatic effects in lung cancer.

Section 4 (lines 419-436)

Previous studies have shown that ROCK1 regulates cancer cell motility, invasion, migration and metastasis [41-43]. Moreover, our current study reveals that cytoplasmic precursor CLU bind ROCK1 to abrogate the interaction between ROCK1 and ERK, leading to suppression of lung cancer migration and invasion. (we revised the description in this part). Consistent with previous reports showing that the treatment of ROCK1 inhibitor Y27632 markedly inhibits the invasion and migration of breast cancer cells in vitro and bone metastasis in vivo [44], our study demonstrate that Y27632 also inhibits the migration of lung cancer cells, especially silenced with CLU expression. As a kinase, ROCK1 importantly induces actin polymerization and increases migratory ability through phosphorylating and activating LIMK and MLC, and phosphorylating actin-depolymerizing protein cofilin, consequently leading to cytoskeletal reorganization [10]. However, our study finds that the binding of CLU to ROCK1 fails to affect the activity or expression of cytoskeleton-reorganizing proteins such as LIMK and cofilin. Notably, it is also reported that ROCK1 can also phosphorylate ERK2 to promote tumor metastasis [12, 45]. Consistent with this report [14], we demonstrate that the binding of CLU to ROCK1 abrogates the interaction between ROCK1 and ERK and thus inhibits ERK activation. Moreover, the inhibition of ROCK1 by treatment with its inhibitor Y27632 remarkably diminishes the upregulation of ERK1/2 phosphorylation induced by CLU knockdown.

Point 5: Conclusion

Can authors improve the section by highlighting the key outcome of the study and the potential impact among the scientific community?

Response 5: We appreciate it very much for this good suggestion, and we have done it according to your ideas. In the revised manuscript, appropriate descriptions have been incorporated into the revised manuscript on lines 438-445 in Page 13.

Cytoplasmic CLU is downregulated in lung cancer and correlates with poor survival. Cytoplasmic precursor CLU inhibits lung cancer metastasis through binding ROCK1 to decrease phosphorylation of ERK1/2. This work reveals a novel function and regulation of cytoplasmic precursor CLU in lung cancer, which might be a potential target for diagnosis and treatment of metastatic lung cancer. Furthermore, our finding extends the current un-derstanding of the kinase activity of ROCK1 without affecting cellular cytoskeleton and identify a novel mechanism underlying ROCK1 mediated ERK phosphorylation and ac-tivation due to CLU loss during lung cancer progression.

Round 2

Reviewer 3 Report

Authors addressed all comments properly. Publication of the paper in its current form is recommended